# Seasonal Characteristics and Particle-size Distributions of Particulate Air Pollutants in Urumqi

**DOI:** 10.3390/ijerph16030396

**Published:** 2019-01-31

**Authors:** Xianyong Meng, Yiping Wu, Zhihua Pan, Hao Wang, Gang Yin, Honggang Zhao

**Affiliations:** 1College of Resources and Environmental Science, China Agricultural University (CAU), Beijing 100193, China; xymeng@cau.edu.cn; 2Department of Civil Engineering, The University of Hong Kong (HKU), Pokfulam 999077, Hong Kong, China; 3Department of Earth & Environmental Science, Xi’an Jiaotong University, Xi’an 710049, China; 4State Key Laboratory of Simulation and Regulation of Water Cycle in River Basin & China Institute of Water Resources and Hydropower Research (IWHR), Beijing 100011, China; wanghao@iwhr.com; 5School of Resources and Environment Science, Xinjiang University (XJU), Urumqi 830046, China; chinayg@xju.edu.cn (G.Y.); xjwzhg@163.com (H.Z.)

**Keywords:** air pollution, air quality monitoring, aerosol, particulate matter, PM concentration, physicochemical property

## Abstract

Urban particulate air pollution is a known cause of adverse human health effects worldwide. Urumqi is a large oasis city in which rapid urbanization has caused a series of eco-environmental problems including serious air pollution, water shortage, dense population, excess energy consumption, and the creation of an urban heat island, among others. Coal is the most important source of energy and air pollutants that are poorly dispersed into the natural surroundings are the main reasons for serious pollution in the Urumqi urban area. Using differential optical absorption spectroscopy (DOAS), aerosol levels were determined using the double optical path method. We found that aerosol concentrations in Urumqi increased rapidly in winter, and that the concentration of fine particles was much higher than that of coarse particles. The background aerosol concentration was highest in winter in the research area, and the air-flow speed had a significant impact on this because high speed surface winds that correspond to high air flows can transport the aerosol to other places. Some of the observed day-to-night differences may be caused by differing wind directions that transport air masses from different emission sources during the day and the night. Daily and seasonal differences in PM_1.0_ concentrations of different grades of polluted air were statistically analyzed using average daily concentration data for particles smaller than 10, 2.5 and 1.0 microns (PM_10_, PM_2.5_ and PM_1.0_), and meteorological observations for Urumqi, Tianshan District in 2010.

## 1. Introduction

Atmospheric aerosol particles are important air pollutants, and the level of aerosol particles is expressed by the mean concentration of fine particulate matter (PM_10_ and PM_2.5_, i.e., the particles are smaller than 10 or 2.5 microns, respectively) [1,2]. The apportionment of the source of airborne particulate matter is important in the field of environmental administration. In many cities, PM_10_ has become the primary pollutant [3]. Since the mid-90s, the Chinese government has adopted increasingly stringent standards for abating emissions and for improving the air quality. As a consequence, a general improvement of air quality has been recorded in the last decade. The development of methods for monitoring and controlling PM_10_ in atmospheric pollution is now a major focus [4]. The influence of aerosols on the city environment impacts in several ways. Firstly, hygroscopic aerosols (such as black carbon and sulfates) in conditions of high relative humidity easily form fog. As air turbidity increases, this fog mixes with dust to become smog, enhancing the scattering and absorption of solar radiation, in turn weakening the influence of solar radiation at ground level, and affecting the regional radiation balance and visibility in the city [5]. Secondly, fine particles can easily adsorb toxic substances (polycyclic aromatic hydrocarbons can easily be adsorbed on the surface of PM_10_). Because of the small size of these aerosol particles, these toxic compounds are able to pass through the respiratory tract, depositing in the alveoli, potentially causing great harm to human health [6]. Because of their different chemical compositions and concentrations, the survival time of particles of various sizes is different [7]. Hygroscopic particles tend to form condensed nuclei. As a consequence, there is an increased possibility of cloud precipitation, resulting in relatively short particle life while the life of non-hygroscopic particles is relatively long. Therefore, it is of great importance to understand the characteristics of atmospheric aerosol pollution and their impact on the environment by studying the particle size, spectral distribution and chemical composition of atmospheric aerosols [8]. In recent years, with the acceleration of climate change and the expansion of urbanization, various environmental problems and their impact on human health have also received the attention of the scientific community [9,10,11,12,13,14,15,16,17,18,19,20,21].

Ambient air quality monitoring stations provide large quantities of temporal data. These data are conveyed to the public as air quality index (AQI) values or other meaningful indices that depend on purpose and time scale, as well as a range of sub-indices that are based on epidemiological studies. The US Environmental Protection Agency (US EPA) initially implemented the use of AQI in 1999, and the methodology was later updated and defined in terms of six key air pollutants: carbon monoxide (CO), nitrogen dioxide (NO_2_), ozone (O_3_), coarse particulate matter (PM_10_), fine particulate matter (PM_2.5_), and sulfur dioxide (SO_2_) [22]. 

AQI values vary from 0 to 500 and their magnitudes are proportional to pollutant concentration in ambient air; greater AQI values have more serious health implications. If the AQI is greater than 100, then the air quality is unhealthy for certain groups. The air pollution categories based on AQI are listed in Appendix A. For each color coded AQI, a unique association exists between the AQI level and its health implications.

## 2. Material and Methods

The sampling station, an air-conditioned container, is located in Urumqi (43.48° N, 87.39° E, Elevation: 935 meters) (Figure 1). According to the January 2013 National Environmental Analysis, Urumqi is among the top ten most air-polluted cities in the world. Heavy haze is extremely common in winter, and frequently affects air traffic. Officials believe that severe winter air pollution in Urumqi is mainly caused by energy intensive industries and outdated coal burning in winter heating systems. According to a report by the Department of Environmental Science and Engineering of Fudan University, the average PM_2.5_ and total suspended particle (TSP) concentrations in the winter of 2007 were 12 times higher than the USA standard for PM_2.5_, and three times the National Ambient Air Quality Standard of China for TSPs. Sulfur dioxide from industrial emissions mixed with local anthropogenic aerosols and transported soil dust from outside the city were the main sources of the high sulfate concentration, one of the main factors responsible for the heavy air pollution over Urumqi. 

Air pollutants emitted from power plants and chemical factories, that are located upwind from the urban area, can affect the air quality over the city area [3]. Because of the southern prevailing wind and the higher elevation of the southern part, it is difficult for pollutions in the urban area to diffuse. Small boilers, without measures for controlling smoke and dust form the main mode of heating in the city’s suburbs. This heating method consumes high amounts of energy and causes serious pollution, with emitted pollutants such as soot and sulfur dioxide increasing greatly during their use. Open-air barbecue stalls are another important source of air pollution in Urumqi, as the city has nearly one million barbecue grills while NO_2_ emitted by motor vehicles reached 0.025 mg·m^−3^ in Urumqi city in 2007. With the rapid growth in the number of vehicles in the downtown area, emissions from motor vehicle exhausts together with sewage from catering services and reduced air flow due to buildings, result in poor air quality. As a result, air pollution is a serious problem in Urumqi today. With the increasing problem of air pollution in Urumqi, the Urumqi government promulgated measures for the prevention and control of air pollution in Urumqi in November 2008 [23], which were used to relieve the pressure of air pollution in Urumqi.

By measuring concentration data and the optical properties of Urumqi air, between December 2009 and October 2010, and by determining the physicochemical properties of aerosol samples over two winters, this research investigated the city’s aerosol pollution characteristics. Using the back-trajectory method, we analyzed the mechanism for how weather conditions influence aerosol pollution in the Urumqi area and investigated the influence of the physicochemical properties of particles with varying diameters on the atmospheric extinction of sunlight. Though similar studies have been reported for other areas [24,25,26,27], such detailed data are quite new for the Urumqi area.

Airborne particles were collected from urban areas of Urumqi using an FA-3 aerosol cascade sampler (Figure 1c) (ZHRX, Beijing, China). The size of the aerosol particles in the samples were measured. Particulate matter includes particles from molecular size to greater than 10 mm in diameter. The particle diameter range was divided into 9 levels. The total measuring time was about one week. After sampling, a high-precision electronic balance was used to weigh the sampling filter membrane. The average mass concentration of aerosol particles in the various size fractions can be calculated as follows [28]:
*ρi* = (*m*_*i*2_ − *m*_*i*1_)/*Vs*

In this formula: *ρi* is the mass concentration of particles of size *i* (g m^−3^), *m_i_*2(g) is the membrane weight after sampling, *m_i_*1(g) is the membrane weight before sampling, and *Vs* is the sampling volume (m^3^) under standard conditions.

Liquid ion chromatography, using a Dixon ICS-3000 ion chromatograph (TechMax, New north city, Taiwan), was used for the determination of water-soluble ions in the various particle size fractions [29].

The diameters of the particles were divided into 9 grades: level 11, 9.0–10 μm; level 2, 5.8–9.0 μm; level 3, 4.7–5.8 μm; level 4, 3.3–4.7 μm; level 5, 2.1–3.3 μm; level 6, 1.1–2.1 μm; level 7, 0.65–1.1 μm; level 8, 0.43–0.65μm; and level 9, submicron. Seven sampling times, of different duration, between December 2010 and March 2011 were employed. Sampling volumes were calculated according to the sampling time and the sampling flow rate. The aerosol mass concentration was obtained by dividing the aerosol weight by the sample volume; the mass concentration is the average concentration of the aerosol over the sampling duration.

## 3. Results and Discussion 

### 3.1. Annual Variations in PM_10_ Concentration

Based on the daily air pollution index (API) promulgated by the China Ministry of Environmental Protection (MEP) in June 2000 [30], air quality can be divided into five air quality grades: excellent, good, light pollution, moderate pollution and severe pollution [30,31]. The mass concentration of PM_10_, which is the main air pollutant during the winter season in Urumqi can be estimated from the API value (see Table 1).

While the daily maximum concentration of PM_10_ in Urumqi was greater than or equal to 600 μg·m^−3^ during the period 2004–2009, the API in Urumqi was set to no greater than 500 (i.e., the maximum), and the minimum daily average concentration of PM_10_ (from 2004–2009) was taken from the real-time air quality index (AQI) monitoring network [32]. The average daily maximum concentration of PM10 dropped to 536 μg·m^−3^ in 2010, which was lower than that for the previous year. This is because Urumqi implemented an energy structure improvement policy aimed at energy conservation, emission reduction and motor vehicle exhaust control. The average annual concentration experienced fluctuating growth during this period, as shown in Table 2. Despite this, air pollution in the city reached the “severe pollution” level. It can be seen that there is an obvious year-to-year variation in the PM_10_ pollution characteristics in Urumqi.

### 3.2. Average Monthly Variations in PM_10_ Concentration

Figure 2 and Figure 3 show the average monthly PM_10_ mass concentration and the percentage of PM_10_ over the period 2004–2010, respectively. The average PM_10_ concentration exceeded national air quality standards in January, February, March, November and December. The highest average value of 338.2 μg·m^−3^ appeared in January and is close to that associated with “moderate pollution”. The second-highest PM value of 293.5 μg·m^−3^corresponded to December while the third, 210.7 μg·m^−3^, occurred in November. The contamination levels corresponded to “light pollution” in March and November, while the percentage of PM_10_ in the primary pollutants over the total number of days was also small over those two months (Figure 3). It is easy to generate an inversion layer at low temperature in winter, in a city that is located in a valley; a deep inversion layer hinders the diffusion and dilution of pollutants [33]. As coal heating is used during the colder months, air pollution emissions also increase significantly during those months. These pollutants include inhalable particulate matter such as PM_10_, while other gaseous pollutants, such as SO_2_ and NO_2_, reduce the number of PM_10_ pollution days. The ground is covered with snow in winter and this causes a significant reduction in dust aerosols [34], As a result, the number of days with PM_10_ as the primary pollutant as a proportion of the total number of days is less than 70%, and from Figure 3 we can observe that this proportion decreased from January to March with March having the least number of days when PM_10_ was the primary pollutant, accounting for only 46.2%.

The PM_10_ mass concentration was lower than that of the national quality standards in the months from April to October, with the minimum value of 60.9 μg·m^−3^ appearing in June, which is far below the national air quality requirements (150 μg·m^−3^). The average mass concentration was lower than 100 μg·m^−3^ in these seven months and the concentrations of PM_10_ met the national air quality standards II, and the air quality was generally good. According to the World Health Organization, the main components of particulate matter are sulfate, nitrate, ammonia, sodium chloride, carbon, mineral dust and water [35]. The proportion of PM_10_ in the primary pollutants exceeded 80% in these seven months, and even exceeded 95% from July to September, which is due to the disappearance of the inversion layer and the increase in rain [36]. The concentration of pollutants in the atmosphere decreased sharply due to erosion and diffusion dilution caused by rain. Soluble SO_2_ and NO_2_ emissions were lower than those observed in winter [37].

### 3.3. Seasonal Variations in PM_10_ Concentration

Seasonal variations need to take into account the division of the seasons for the city; namely spring (March, April and May), summer (June, July and August), autumn (September, October and November), and winter (December, January and February). Seasonal average mass concentrations of PM_10_ are displayed in Figure 4, which shows that the air quality is the clearest in summer when there is a minimum amount of inhalable particulate matter in the atmosphere. The seasonal average PM_10_ mass concentrations were less than 100 μg·m^−3^ and air quality over the non-heating period was good, but the PM_10_ mass concentration over summer shows a fluctuating, but increasing trend since 2005. As the number of motor vehicles increases year-by-year, so does the amount of inhalable particulate matter discharged into the air. Changes in meteorological conditions are also important factors that affect annual changes in pollutant levels [38]. Due to substantial increases in pollutant emissions and enduring low-altitude inversion layers over the city in winter, the air quality is the worst in winter. Average PM_10_ mass concentration in each season in Urumqi reached a maximum value of 358.9 μg·m^−3^ in the winter of 2005. Since 2005, the concentration of PM_10_ in the winter has been decreasing year-by-year, and has reached 255.2 μg·m^−3^; pollutant discharge has been effectively controlled through the introduction of central heating [39].

The local and regional distribution of pollutants is significantly influenced by weather patterns and their variability, along with spatial emission patterns. Heating periods decrease when there are warm winters. As less PM_10_ is discharged, less CO_2_ and SO_2_ are also produced, promoting improved environmental air quality. The data show that the winter concentration of PM_10_ has decreased year by year since 2006 (Figure 4); the average mass concentration of PM_10_ was 113.1 µg·m^−3^ in spring and 122.6 µg·m^−3^ in autumn, as shown in Figure 5. Periods of heating in spring are less than in autumn. We propose that the day-by-day accumulation of PM_10_, from its lowest value to its peak value, dropping back to the lowest value, is an environmental pollution process. The temperature inversion is strongest in winter and weakest in summer [40].

### 3.4. Diurnal Variations in PM_10_, PM_2.5_ and PM_1.0_ Mass Concentrations

PM mass concentrations from December 2009 to November 2010 were analyzed and the diurnal variations in the mass concentration of PM_10_, PM_2.5_ and PM_1.0_ are displayed in Figure 5. There are five fluctuations in PM mass concentration every day. Maximum concentrations occurred at 12 am, 5 am, 10 am, 3 pm and 8 pm while minimum values were observed at 3 am, 8 am, 12 pm, 4 pm, 9 pm and 10 pm. The minimum value observed corresponded to 3 am, while the maximum value occurred at 8 pm. The main conclusions are as follows: PM mass concentrations in Urumqi increase from 3 am to 8 pm, but also fluctuate; the air pollution problem, especially atmospheric air pollution mainly results from mass emissions from vehicles, and is becoming more serious with the continual increase in the number of automobiles, especially in winter. A steady stream of city traffic significantly contributes to air pollution in the daytime. Over the city area, the development of an inverse temperature layer is strong and thick and particulate air pollution is not easily dispersed, fluctuations in PM concentrations increase in the daytime [41].

According to the data accumulated by monitoring and analysis of atmospheric aerosols at the surface boundary layer over Urumqi city from 2009 to 2010, PM mass concentrations fluctuate according to changes in environmental conditions during the day. The formation of radiation inversion in the surface layer was largely due to the cooling effect of long-wave radiation at night. The average wind speed was low during the early morning hours in 2009 and 2010, therefore, it is difficult to eliminate air pollution [42]; consequently, peak aerosol concentrations were observed at 10 am. 

There exists a clear and positive feedback mechanism between the intensity of the atmospheric inversion and pollutants. With ever-increasing temperatures, the inversion layer gradually diminishes. Upward vertical turbulent heat flux gradually increases, which improves the atmospheric diffusion process of these particulates. While the results show that the diurnal PM mass concentrations vary greatly, there is an obvious decreasing trend in which PM concentrations decrease due to enhanced air flow during the daytime. Automobile exhaust emissions are an important source of fine PM_2.5_ and PM_10_ particulate matter; thus, decreases in vehicle traffic clearly result in reduced PM concentrations from vehicle exhaust emissions.

Figure 6 reports data collected during diurnal cycles. As was observed for the seasonal patterns, daily cycles are also the result of the interplay between the intensity of various sources such as photochemical processes and meteorological factors. Although minor changes due to specific local conditions are observed, almost all sites exhibit similar daily and weekly cycles. CO and nitrogen oxides show typical daily cycles linked to road traffic, with two daily maxima corresponding to the morning and evening rush hours (7–9 am and 6–8 pm). The morning and evening maxima are split by a period of low emission, which is assumed to be the result of: (i) lower emissions (less traffic); (ii) larger availability of ozone driven by the daylight photolysis of NO_2_ and the oxidation of volatile organic compounds (VOC); and CO (iii) higher convective activity leading to a deeper mixed layer, which enhances atmospheric mixing. Weekly patterns are also linked to road traffic; generally, average levels increase from Mondays to Thursdays, while a significant drop is observed over the weekends, when road traffic reaches minimum volumes and heavy-duty vehicles over 7.5 tons are subject to restrictions.

Based on the Urumqi air quality index map [43], ozone and total oxidants (OX) show daily peaks in the mid-afternoon, i.e., the hours that experience higher solar radiation levels, and lower levels are experienced between 6 and 9 am local time (daylight-saving time corrected in summer). These patterns are also enhanced in summer due to generally higher levels of solar radiation. It is evident that daily peaks of OX are delayed by 2–3 h with respect to ozone, corresponding to increases in NO to NO_2_ oxidation and primary NO_2_ emissions during the evening rush hours [44]. This “flatter” pattern is likely to be related to the lack of anthropogenic sources of freshly emitted ozone precursors, and the presence of higher levels of biogenic ozone-precursors, which do not follow anthropogenic cycling [45]. However, the levels of ozone are also known to be strongly affected by the transport of polluted air masses by local wind systems [46]; nocturnal dry deposition is also less effective in city sites.

Generally, PM_10_ exhibits higher concentrations overnight with clear minima in the early afternoon. This pattern is consistent with the diurnal dynamics of the mixing layer [47]). However, a secondary cause may be related to the volatilization of the more volatile aerosol compounds (e.g., nitrate) during the early afternoon, i.e., when the air temperature is higher and relative humidity is lower. Minor PM_10_ concentration peaks are observed just before noon and can be caused by very different mission scenarios. The interpretation of these observations is not clear and may be related to the local characteristics of the observation sites.

Figure 6 shows the diurnal changes of PM_2.5_/PM_10_, PM_1.0_/PM_10_, PM_1.0_/PM_2.5_ mass concentrations over the four seasons in Urumqi. We found three maxima and three minima during all four seasons. Comparing the ratios of different seasons suggested that the three ratios in winter are higher than those in other seasons, which reflects the effect of the winter inversion layer on the atmosphere. For Winter (PM_2.5_/PM_10_), the maximum value appeared at 2 pm, 9 am, 10 pm and 3 am, the minimum value appeared at 6 pm, 8 pm, 11 am and 4 am. The maximum value of PM mass concentrations ratio appeared at 3 am, 12 am and 10 pm for Winter (PM_1.0_/PM_10_), the minimum value appeared at 9 pm. Unlike the above two ratios, the minimum value appeared at 5–7 am for Winter (PM_1.0_/PM_25_), the ratios show a smaller range of variations at other times. For spring, the minimum value appeared at 3–5 am for Spring (PM_2.5_/PM_10_), spring (PM_1.0_/PM_10_) and (PM_1.0_/PM_2.5_). The maximum value appeared at 11 am, 7 pm and 4 pm, respectively. In summer, the maximum values appeared at 6 am and 0 am, while the second maximum value appeared at 10 pm. With the disappearance of the inversion layer and reduction of traffic flow, the minimum of the PM mass concentration appeared at noon (from 11 am to 7 pm). With enhancements in atmospheric convection, the lowest values of particle concentration appeared at 6pm. With increases in traffic flow, the night time radiation inversion layer and automobile exhaust emissions, nitrogen dioxide, carbon monoxide and particulates increase dramatically, with the peak value of particle concentrations appearing at 10 pm. For Autumn (PM_1.0_/PM_2.5_), the maximum value appeared at 10 am in the morning, the next highest at 8 pm, with the third peak appearing at 0 am. The minimum value appeared at 2 pm, the next lowest at 5 pm, and the third lowest at 11 pm–3 am. For Autumn (PM_2.5_/PM_10_) and (PM_1.0_/PM_10_), the maximum value appeared at 5–7 am, the second highest at 10 pm with the third peak appearing at 10 pm and 0 am. The difference in the peak values was small and seasonal. Similar to the observations for autumn and summer, the maximum PM concentration ratio value appeared in the early morning for both Autumn (PM_2.5_/PM_10_) and (PM_1.0_/PM_10_). 

### 3.5. Seasonal and Monthly Average Ratios of PM_2.5_/PM_10_, PM_1.0_/PM_10_ and PM_1.0_/PM_2.5_


The ratio of PM_2.5_/PM_10_, PM_1.0_/PM_10_ and PM_1.0_/PM_2.5_ is a useful characterization tool that assists in the spatial and seasonal identification of the dominant aerosol types [48]). High ratios indicate the dominance of anthropogenic aerosols whereas low ratios indicate the dominance of dust aerosols. The dimensionless seasonal PM_2.5_/PM_10_ concentration ratios were calculated for winter, spring, summer and autumn to identify the dominant particulate matter pollution in Urumqi. Figure 7 and Figure 8 show the seasonal and monthly average ratios of PM_2.5_/PM_10_, PM_1.0_/PM_10_ and PM_1.0_/PM_2.5_.

The figures and data indicate that the PM_10_ concentrations are higher in winter in comparison to summer. The seasonal and 12-monthly variations in PM_2.5_/PM_10_, PM_1.0_/PM_10_ and PM_1.0_/PM_2.5_ in Urumqi show similar trends. However, the PM_1.0_/PM_2.5_ ratios were observed to be highest in winter indicating that fine particles arise predominately from anthropogenic sources, such as industrial activity and heavy traffic, during the winter period, in agreement with literature that confirms the prevalence of PM_2.5_ in the immediate proximity to industrial areas [49].

### 3.6. Distribution Characteristics of Aerosol Particles within Each Size Interval in Urumqi

Airborne particles were collected from urban areas of Urumqi using an FA-3 aerosol cascade sampler in winter 2010, as described in Section 2 (Materials and Methods). The size fractions of the aerosol particles in the various samples were measured.

Figure 9 shows the average aerosol concentration distribution across the nine particle diameter grades (Table 3) in winter. Because the 2.5 μm filter was not available for the sampler, we took the aerodynamic equivalent diameter of 2.1 μm (i.e., the 4^th^ grade) to define the fine particle. The coarse particles are mostly dust and coal ash with a size of 2.1–10 μm, accounting for nearly 81% in terms of the size range, However, aerosol mass concentration account for only 42.5% of the total mass concentration of PM_10_. This suggested that the contribution of coarse particles (grade 0 to grade 4) was relatively low with grade 2 (4.7–5.8 μm) being the lowest (12.8 ± 9.1 μg·m^−3^). It was clear that the mass concentration of particles falling in grade 5 is the highest (23.0 ± 10.6 μg·m^−3^), followed by grade 6. As a whole, Figure 9 indicates that the particulate pollution was dominated by fine particles, a typical aerosol pollution type. 

## 4. Conclusions

Based on the data for aerosol concentration and optical characteristics measured between December 2009 and March 2011 and the physical and chemical characteristics of aerosol samples collected in winter, the pollution characteristics of local aerosols were analyzed, and the influence of meteorological conditions on local pollution was analyzed by means of posterior trajectory clustering analysis. The effects of physicochemical properties of aerosol particles of different particle sizes on atmospheric extinction were also discussed. The results are as follows.

In spring, the aerosol concentration decreases as a result of air flow from the desert region, resulting in aerosol particles becoming coarser. In summer, clean air flow, through wind speed, influences aerosol concentration. In autumn, the aerosol mass concentration is greater than in summer and the air flowing from the desert region changes the aerosol concentration distribution, but the aerosol concentration does not change much, which indicates that the dust aerosol transported to this area is limited.

Seasonal pattern analyses reveal that PM_1.0_, PM_2.5_ and PM_10_ show significantly higher levels during the colder months, with minimum levels in summer. This pattern is mainly attributed to lower mixing-layer heights, limited potential for oxidation, and lower emissions from domestic heating. The volatilization of semi-volatile aerosol compounds during the warmer seasons is another reason for this PM behavior. On the contrary, ozone exhibits opposite seasonality, with maxima in the summer due to its increased generation through photochemical processes.

Because of the instability of the observation instrument, resulting in the lack or distortion of the data, there may be some deviation in the results of this study We will minimize the uncertainty caused by the observation instrument in the future, so as to obtain more accurate real analysis results.

## Figures and Tables

**Figure 1 ijerph-16-00396-f001:**
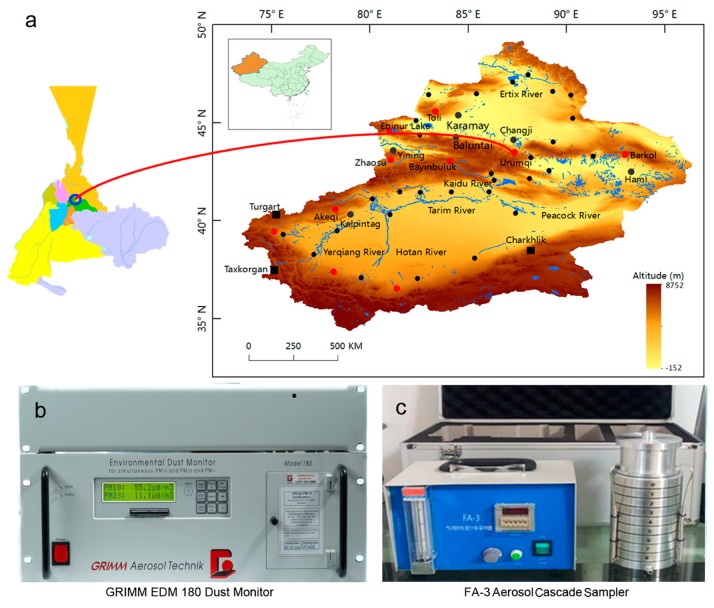
The geographical location of the study area and the observation instruments placed in the study area. (**a**). Location of the Urumqi city (Xinjiang, China) sampling station. (**b**). The EDM 180 (GRIMM, Ainring, Germany), provide simultaneous measurement of PM_1.0_, PM_10_ and PM_2.5_ with low maintenance, fast response (6 sec), real-time monitoring, insensitivity to vibration (mobile) and lowest ownership cost for this research. (**c**). The FA-3 aerosol cascade sampler.

**Figure 2 ijerph-16-00396-f002:**
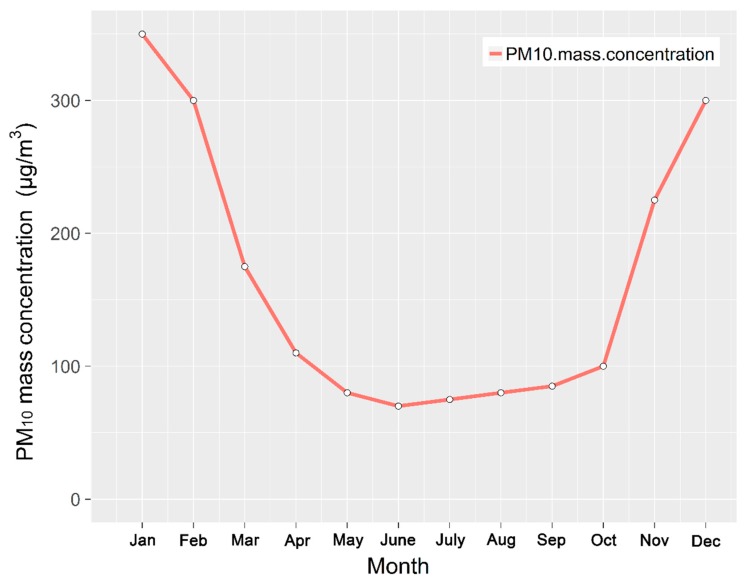
Average monthly PM_10_ mass concentration over the period 2004–2010.

**Figure 3 ijerph-16-00396-f003:**
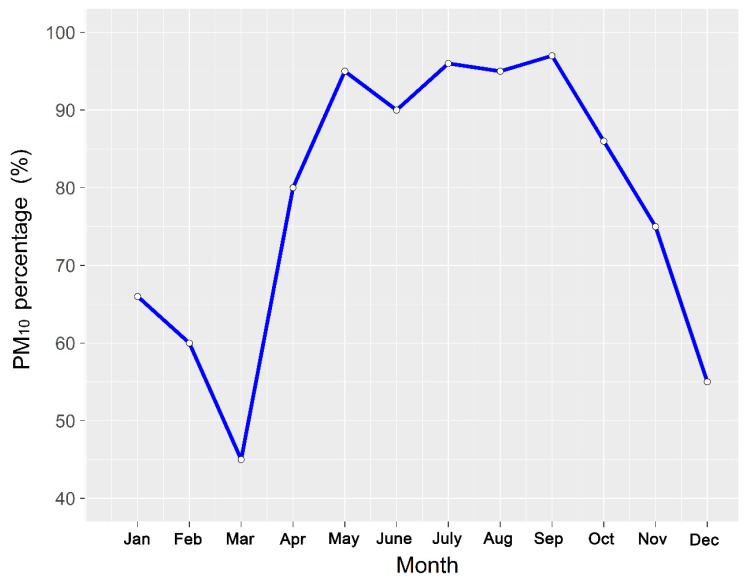
Average monthly percentage of PM_10_ over the period 2004–2010.

**Figure 4 ijerph-16-00396-f004:**
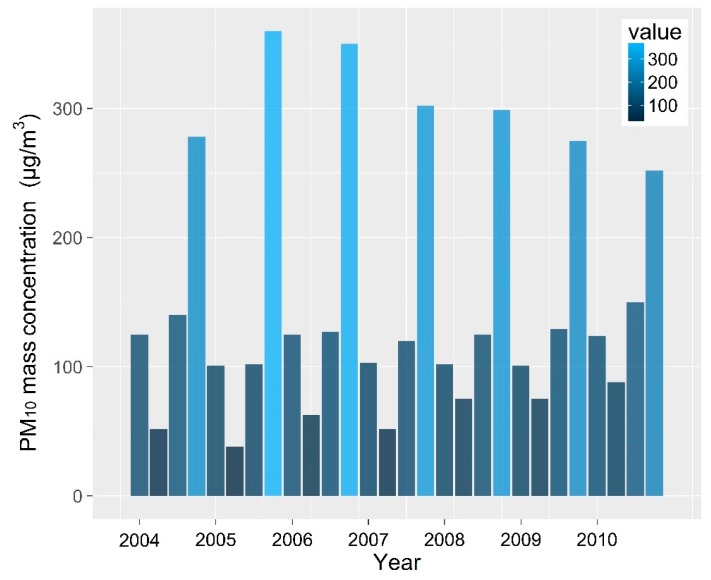
Yearly average mass concentration of PM_10_ (2004–2010).

**Figure 5 ijerph-16-00396-f005:**
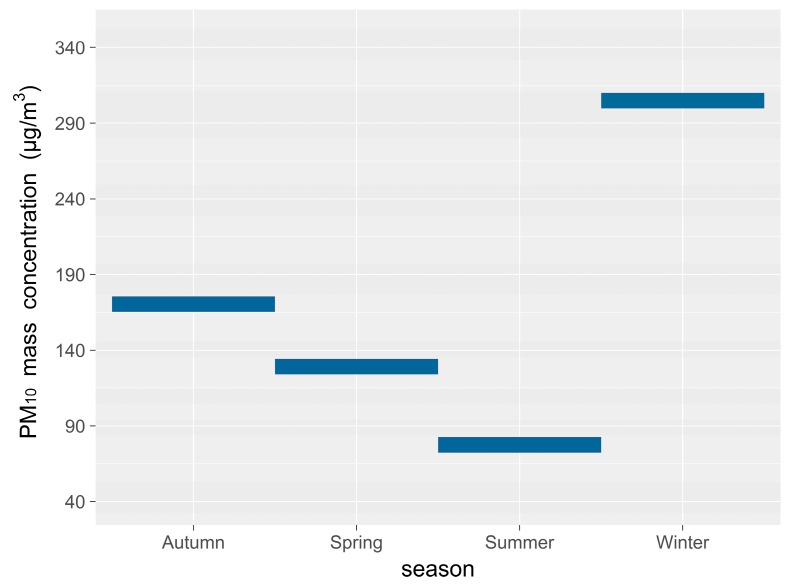
Seasonal average mass concentration of PM_10._

**Figure 6 ijerph-16-00396-f006:**
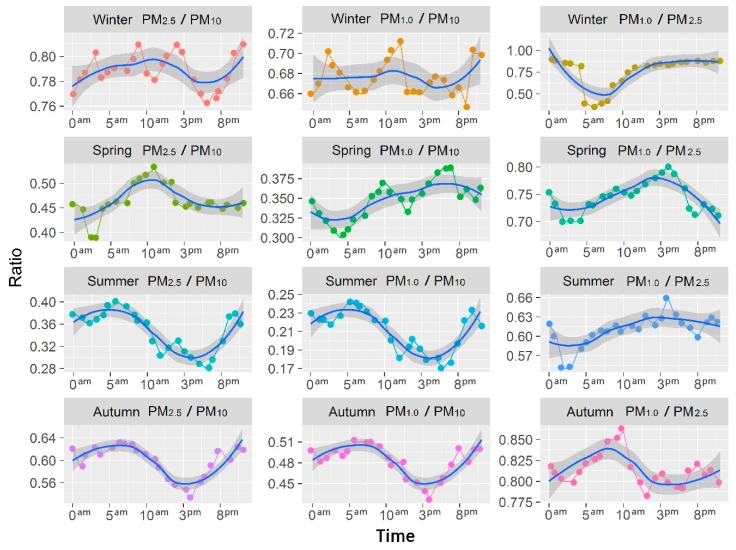
Diurnal changes of PM_2.5_/PM_10_, PM_1.0_/PM_10_, PM_1.0_/PM_2.5_ mass concentrations over the four seasons.

**Figure 7 ijerph-16-00396-f007:**
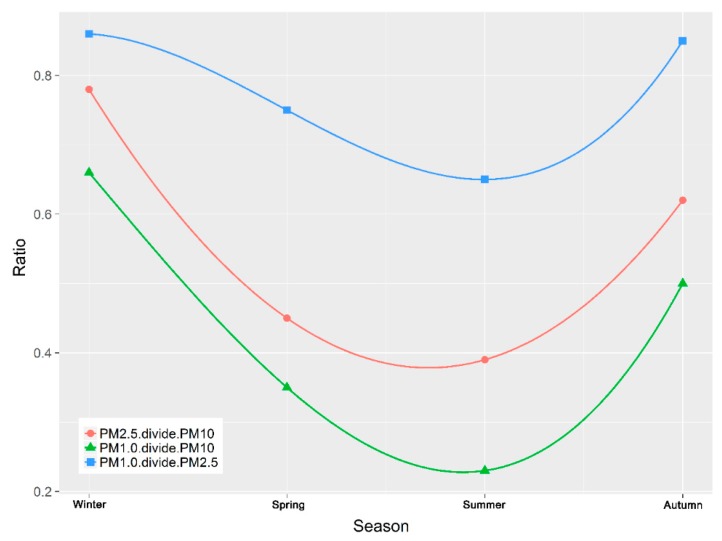
Seasonal ratios of PM_2.5_/PM_10_, PM_1.0_/PM_10_, PM_1.0_/PM_2.5._

**Figure 8 ijerph-16-00396-f008:**
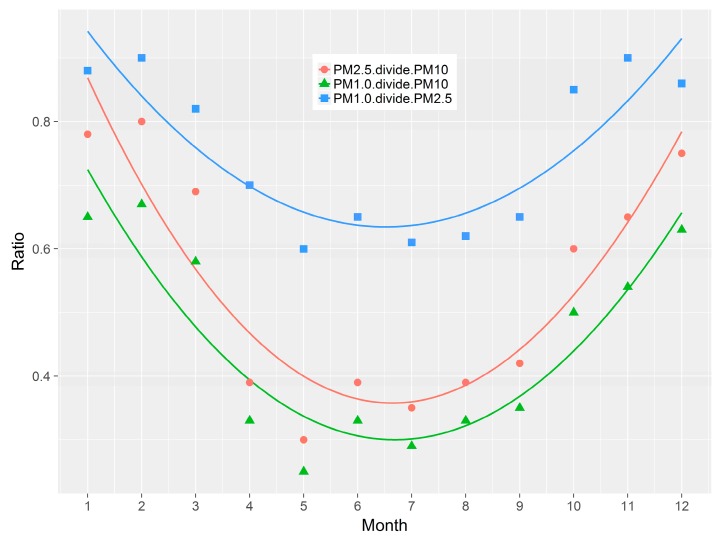
Monthly ratios of PM_2.5_/PM_10_, PM_1.0_/PM_10_, and PM_1.0_/PM_2.5._

**Figure 9 ijerph-16-00396-f009:**
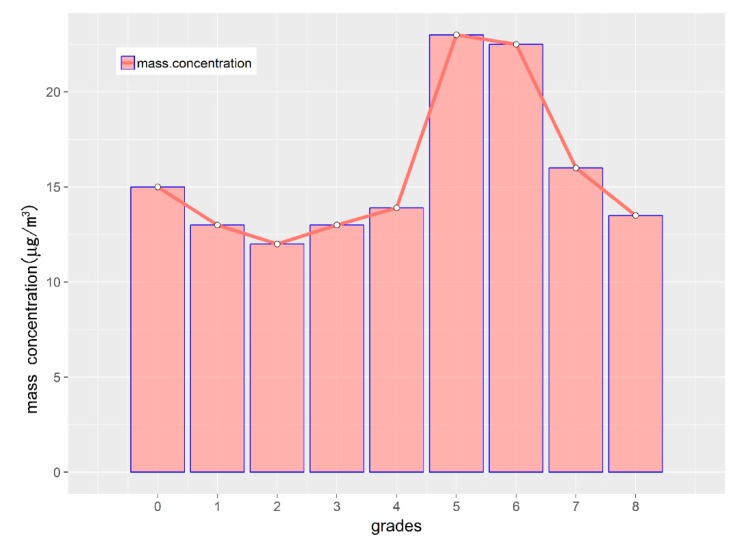
The mass concentration distribution across the nine particle diameter grades in winter.

**Table 1 ijerph-16-00396-t001:** Formulas for transforming API into PM_10_ mass concentrations.

API	<50	51–100	101–200	201–300	301–400	401–500
Air quality grade	excellent	good	light pollution	moderate pollution	severe pollution	severe pollution
PM10 concentration	C = API R^2^ = 1.0	C = 2API − 50 R^2^ = 1.0	C = 2API − 50 R^2^ = 0.98	C = 0.7API + 210 R^2^ = 0.99	C = 0.8API + 180 R^2^ = 0.69	C = API + 100 R^2^ = 1.0

**Table 2 ijerph-16-00396-t002:** Annual variations in PM_10_ mass concentration.

Year	Percentage of Total Number of Days *(%)	Average Daily Minimum Concentration (μg·m^−3^)	Average Daily Maximum Concentration (μg·m^−3^)	Average Annual Concentration (μg·m^−3^)	Standard Deviation (μg·m^−3^)
2004	73.2%	14	600	124.6	123.3
2005	69.9%	15	600	119.3	140.6
2006	87.9%	24	600	137.9	127.8
2007	89.0%	24	600	141.3	124.7
2008	83.1%	27	600	148.6	117.4
2009	89.3%	18	600	144.3	116.9
2010	83.5%	25	536	140.5	84.4

* Number of days of major air pollutant per year as a proportion of total days.

**Table 3 ijerph-16-00396-t003:** Particle diameter grade.

Grades	0	1	2	3	4	5	6	7	8
Range	9.0–10 μm	5.8–9.0 μm	4.7–5.8 μm	3.3–4.7 μm	2.1–3.3 μm	1.1–2.1 μm	0.7–1.1 μm	0.43–0.65 μm	100–1000 nm

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
