# Peer review of "Seasonal Characteristics and Particle-size Distributions of Particulate Air Pollutants in Urumqi"

_ijerph, 2019, doi:10.3390/ijerph16030396_

Round 1

Reviewer 1 Report

This study investigated DOAS method to measure the level of particulate maters with different size ranges at different times and durations in Urumqi in 2010. The authors did a very goof job in collecting, processing and analyzing them. My main concern about this paper is presenting the data and discussing them in a clear way. Currently in several points I found the discussion very confusing. In some cases the link between the discussion and the presented data is very vague. Sometimes the authors discuss some data on other pollutants such as ozone that they need either provide the data for them or cite a reference. In general, I believe this the results and discussion section should be extensively revised. Moreover, significant revision is needed in terms of English language. After completing the revisions, I recommend the paper for publication. I have my detail comments as follow

Line 18: What does “high population concentration” mean here?

Line 30: Why the data for 2010 were selected? Why not more recent data sets?

Line 36: Please check the grammar and rewrite: “Atmospheric aerosol particles are important components of air pollution and air quality is expressed by the mean concentration of fine particulate matter”.

Line 55: should be “tend to form”

Line 91: What do you mean by “dissipation” is leading to dilution? What is the difference between “dissipation” and “dispersion”?

Line 106: Here you mentioned the objective of the study. Is there any previous works conducting the same measurements? If there is not similar study for this city, there should be the similar measurements for other locations. Please cite them to better highlight the novelty of your study.

Line 128: “Based on the daily air pollution index (API)promulgated by the State Environmental Protection Administration in June 2000, air quality can be divided into five air quality grades: excellent, good, light pollution, moderate pollution and severe pollution.” Please cite the relevant reference.

Table 1: Please mention the concentration unit for PM10? If you derived the correlations between API and concentration, you need to discuss how you did that? Otherwise, please cite a relevant reference.

Line 135: “600μg·m-3during the period2004 to 2009, the average daily concentration of PM10 dropped to 536μg·m” I believe you mean the average daily maximum concentration is 536? “maximum” is omitted.

Table 2: Since it was previously mentioned that the measurements in this study were conducted in 2010, the data in Table 2 are brought from other sources. In this case, they should be cited. The maximum concentration for all the years except for the measured year are the same and 600. How can you justify the decrease happened only in 2010? Plus, is ca. 11% decrease is significant enough to discuss (from 600 to 536)? Can’t it be attributed to the experimental errors? Also why no discussion on the other parameters such as annual average concentration is provided?

Line 146: “Figures 2 and 3 show the average monthly PM10 mass concentration over the period 2004 – 2010.” Figure 3 is not concentration.

Figure 2 and 3: I strongly recommend to use the name of months for x-axis instead of number to better clarify the discussion.

Line 151: “the proportion of PM10in the primary pollutants over the total number of days was also small over those two months.” Where are these proportions provided?

Line 157: “The ground is covered with snow in winter and this causes a significant reduction in dust aerosols” How is it reflected in your data?

Line 162: “The average mass concentration was lower than 100μg·m-3in these seven month and the concentrations of PM10, SO2 and NO2 met the national  air quality standards II, and the air quality was generally good.” But you provided no data for NO2 and SO2.

Line 166: “The proportion of PM10in the primary pollutants exceeded 80% in these seven months, and even exceeded 95% from July to September, which is due to the disappearance of the inversion layer and the increase in rain.” How your data support this? You just showed PM10 data where are the data for other primary pollutants?

Line 178: “100 g.m-3” the unit should be in micro gram per cubic meter.

Line 185: The first PM10 should be deleted.

Line 192: Please be consistent in using micro sign in the units either use “u” or “μ”.

Line 191: “The data show that the concentration of PM10 decreased year by year; the average mass concentration of PM10was. 1ug.m-3in spring and 122. 6μg.m-3in autumn, as shown in Figure 4.”

According to what graph you noted that “the concentration of PM10 decreased year by year”?

Line 221: “The results suggest that there is a close connection between the microphysical structure of the PM and the degree of air pollution and a long-lived inversion has an effect on the former.” The sentence is unclear. Please rewrite it.

Line 245: “Ozone and total oxidants (OX) show daily peaks in the mid-afternoon, i.e. the hours that experience higher solar radiation levels, and lower levels are experienced between 6 and 9 am local  time (daylight-saving time corrected in summer).” Where are these data (ozone and OX) you discussed?

x-axis in Fig. 5 should be changed into am/pm to be better linked with the provided discussion. Moreover, the legend at the top part of each figure is very confusing. For instance why PM1.0PM10? Is it a range of PM size between 1 to 10 um? If so you should clearly mention it in the discussion.  

Line 263: “The PM concentration displayed three maxima and three minima during autumn, the maximum value appeared at 10pm in the evening, the next highest at 9 – 10 am, with the third peak appearing  at 2pm. The minimum value appeared at 5 am, the next lowest at6pm, and the third lowest at 12 – 1  pm.” There are three figures for autumn. You should specifically mention you are talking about what figure before referring to the maximum values.

Line 265: “The minimum value appeared at 5 am, the next lowest at6pm, and the third lowest at 12 – 1 265 pm. The difference in the peak values was small and seasonal” I also could not spot these values on the figures.

In general, please provide a more clear discussion for Fig. 5 as it has interesting information. Please discuss the trends by referring to the specific figures not a season in general.  

Line 303: Please prepare a table showing eacg grade corresponds to what size range. You may also use x-axis of the Figure 8.

Line 305: “size”

What do you mean by 42.5% of PM10? I think instead of PM10 it should be the amount of particles in all ranges. Then the percentage would be representative.

Line 307: If you have the mass fractions, it is nice to include them in a figure for each grade.

Line 313: This is the place for conclusion or summary not discussion. Please modify the section and make more concise.

Author Response

Response to Reviewer 1 Comments

Point 1: What does “high population concentration” mean here?

Response 1: air pollutants

Point 2: Why the data for 2010 were selected? Why not more recent data sets?

Response 2: Because Urumqi's pollution data acquisition is difficult, and the latest data is generally not accessible, we can only select the best data available or obtained for analysis, in order to obtain the most accurate analysis results. Most importantly, the data in recent years mutated in 2010, and we think this change needs to be reflected, and for the latest data, we will give the next study.

Point 3: Please check the grammar and rewrite: “Atmospheric aerosol particles are important components of air pollution and air quality is expressed by the mean concentration of fine particulate matter”.

Response 3: Changed.

Point 4: Should be “tend to form”

Response 4: Changed.

Point 5: What do you mean by “dissipation” is leading to dilution? What is the difference between “dissipation” and “dispersion”?

Response 5: Changed.

Point 6: Here you mentioned the objective of the study. Is there any previous works conducting the same measurements? If there is not similar study for this city, there should be the similar measurements for other locations. Please cite them to better highlight the novelty of your study.

Response 6: We have joined similar studies and presented the novelty of our study.

Point 7: Based on the daily air pollution index (API) promulgated by the State Environmental Protection Administration in June 2000, air quality can be divided into five air quality grades: excellent, good, light pollution, moderate pollution and severe pollution.” Please cite the relevant reference.

Response 7: We have added the ref:

Liu, J., & Zhang, W. (2015). The influence of the environment and clothing on human exposure to ultraviolet light. PloS one, 10(4): e0124758.

MEP. Technical Regulation on Ambient Air Quality Index(on trial): http:// kjs.mee.gov.cn/ hjbhbz /bzwb/ jcffbz/ 201203/ W020120410332725219541.pdf (accessed on 1 Jan 2016).

Point 8: Please mention the concentration unit for PM10? If you derived the correlations between API and concentration, you need to discuss how you did that? Otherwise, please cite a relevant reference.

Response 8:  We have added the ref:

MEE.Technical Regulation on Ambient Air Quality Index(on trial): http:// kjs.mee.gov.cn/ hjbhbz /bzwb/ jcffbz/ 201203/ W020120410332725219541.pdf (accessed on 1 Jan 2016).

Point 9: “600μg·m-3during the period2004 to 2009, the average daily concentration of PM10 dropped to 536μg·m” I believe you mean the average daily maximum concentration is 536? “maximum” is omitted.

Response 9: We have modified it to the average daily maximum concentration.

Point 10: “600μg·m-3during the period2004 to 2009, the average daily concentration of PM10 dropped to 536μg·m” I believe you mean the average daily maximum concentration is 536? “maximum” is omitted.

Response 10: We have modified it to the average daily maximum concentration.

Point 11: Table 2: Since it was previously mentioned that the measurements in this study were conducted in 2010, the data in Table 2 are brought from other sources. In this case, they should be cited. The maximum concentration for all the years except for the measured year are the same and 600. How can you justify the decrease happened only in 2010? Plus, is ca. 11% decrease is significant enough to discuss (from 600 to 536)? Can’t it be attributed to the experimental errors? Also why no discussion on the other parameters such as annual average concentration is provided?

Response 11: During this period, the API in Urumqi was set to no greater than 500 (i.e., the maximum).

 The average daily maximum concentration of PM10 dropped to 536μg·m-3 in 2010, which was lower than that for the previous year, this is because Urumqi implemented the energy structure improvement policy, aiming for energy conservation and emission reduction and motor vehicle exhaust control. The average annual concentration experienced fluctuating growth during this period, as shown in Table 2. Despite this, air pollution in the city had reached the "severe pollution" level. It can be seen that there is an obvious year-to-year variation in the PM10 pollution characteristics in Urumqi.

Point 12: “Figures 2 and 3 show the average monthly PM10 mass concentration over the period 2004 – 2010.” Figure 3 is not concentration.

Response 12: Changed.

Point 13: Figure 2 and 3: I strongly recommend to use the name of months for x-axis instead of number to better clarify the discussion.

Response 13: Changed.

Point 14: Line 151: “the proportion of PM10in the primary pollutants over the total number of days was also small over those two months.” Where are these proportions provided?

Response 14: We have added (Figure 3) callout after the sentence.

Point 15:  Line 157: “The ground is covered with snow in winter and this causes a significant reduction in dust aerosols” How is it reflected in your data?

Response 15: As a result, the ratio the number of days with PM10 as the primary pollutant to the total number of days is less than 70%, and from Figure 3 we found that this proportion decreased from January to March with March being the least number of days when PM10 being the primary pollutant, accounting for only 46.2%.

Point 16: Line 162: “The average mass concentration was lower than 100μg·m-3in these seven month and the concentrations of PM10, SO2 and NO2 met the national air quality standards II, and the air quality was generally good.” But you provided no data for NO2 and SO2.

Response 16: We deleted the NO2 and SO2.

Point 17:  Line 166: “The proportion of PM10in the primary pollutants exceeded 80% in these seven months, and even exceeded 95% from July to September, which is due to the disappearance of the inversion layer and the increase in rain.” How your data support this? You just showed PM10 data where are the data for other primary pollutants?

Response 17: We have added the citation literature to support this section: Mamtimin, B., & Meixner, F. X. (2011). Air pollution and meteorological processes in the growing dryland city of Urumqi (Xinjiang, China). Science of The Total Environment, 409(7), 1277-1290.

Point 18: Line 178: “100 g.m-3” the unit should be in micro gram per cubic meter.

Response 18: Changed.

Point 19: Line 185: The first PM10 should be deleted.

Response 19: Deleted.

Point 20: Line 192: Please be consistent in using micro sign in the units either use “u” or “μ”.

Changed.

Response 20: Deleted.

Point 21: Line 191: “The data show that the concentration of PM10 decreased year by year; the average mass concentration of PM10was. 1ug.m-3in spring and 122. 6μg.m-3in autumn, as shown in Figure 4.”

Response 21: We lost Figure 4 and rejoined Figure 4.

Point 22: According to what graph you noted that “the concentration of PM10 decreased year by year”?

Response 22: The data show that the winter concentration of PM10 decreased year by year since 2006.

Point 23: Line 221: “The results suggest that there is a close connection between the microphysical structure of the PM and the degree of air pollution and a long-lived inversion has an effect on the former.” The sentence is unclear. Please rewrite it.

Response 23: Changed.

Point 24: Line 245: “Ozone and total oxidants (OX) show daily peaks in the mid-afternoon, i.e. the hours that experience higher solar radiation levels, and lower levels are experienced between 6 and 9 am local time (daylight-saving time corrected in summer).” Where are these data (ozone and OX) you discussed?

Response 24: Based on the monitoring data (http://aqicn.org/city/wulumuqi/cn/).

Point 25: x-axis in Fig. 5 should be changed into am/pm to be better linked with the provided discussion. Moreover, the legend at the top part of each figure is very confusing. For instance why PM1.0PM10? Is it a range of PM size between 1 to 10 um? If so you should clearly mention it in the discussion.

Response 25: We have modified the picture as recommended by the reviewer, in fact, it should be PM1.0/PM10.

Point 26 Line 263: “The PM concentration displayed three maxima and three minima during autumn, the maximum value appeared at 10pm in the evening, the next highest at 9 – 10 am, with the third peak appearing at 2pm. The minimum value appeared at 5 am, the next lowest at6pm, and the third lowest at 12 – 1 pm.” There are three figures for autumn. You should specifically mention you are talking about what figure before referring to the maximum values.

Response 26: We have modified the picture as recommended by the reviewer, in fact, it should be PM1.0/PM10.

Point 27 Line 265: “The minimum value appeared at 5 am, the next lowest at6pm, and the third lowest at 12 – 1 265 pm. The difference in the peak values was small and seasonal” I also could not spot these values on the figures.

Response 27: Changed.

Point 28 Line 265: In general, please provide a clearer discussion for Fig. 5 as it has interesting information. Please discuss the trends by referring to the specific figures not a season in general. 

Response 28: We've added more details in the original paragraph.

Point 29 Line 303: Please prepare a table showing eacg grade corresponds to what size range. You may also use x-axis of the Figure 8.

Response 29: Changed.

Point 30 Line 305: “size”

Response 30: Changed.

Point 31: What do you mean by 42.5% of PM10? I think instead of PM10 it should be the amount of particles in all ranges. Then the percentage would be representative.

Response 31: However, aerosol mass concentration account for only 42.5% of the total mass concentration of PM10. but its mass was 42.5% of that of PM10.

Point 32: Line 307: If you have the mass fractions, it is nice to include them in a figure for each grade.

Response 32: We've given the Table 3.

Point 33: Line 313: This is the place for conclusion or summary not discussion. Please modify the section and make more concise.

Response 33: Changed.

Reviewer 2 Report

The manuscript “Seasonal Characteristics and Particle-size Distributions of Particulate Air Pollutants in Urumqi” by Meng et al. to evaluate the seasonal variability of air pollutants in Urumqi, China (which has an estimated population of 3.5 million in 2015, being the largest city in China’s western interior). A study focusing on the seasonal particulate air pollution on a city as Urumqi would be very interesting. However, the manuscript is full of lack of crucial information to understand what was studied and how it was. Moreover, the manuscript is not consistent at all with misleading information throughout and results’ analysis without being based in any available data.

The references are not updated with a total of 26 with only 1 reference from the last 7 years – 17 from 2000-2009 and 8 from 1999 or before. This shows how superficial the authors worked in this paper without providing a review of the latest work done in the study area – at least that should been done. For instance, with a short (1 min max) research, I found the following article:

Song et al. (2015) A Multiyear Assessment of Air Quality Benefits from China's Emerging Shale Gas Revolution: Urumqi as a Case Study. ENVIRONMENTAL SCIENCE & TECHNOLOGY, Volume: 49  Issue: 4  Pages: 2066-2072. DOI: 10.1021/es5050024

Taking in account all the problems found in the manuscript, I believe it does not gather the quality for publication. I believe that the authors should review the paper to improve it and resubmit again but, at this moment and for the journal, my suggestion is to reject as it stands or suggest to resubmit after a revision from the authors to be evaluate a new time by reviewers.  

Specific comments:

-          The authors at some moments specify some pollutants and some techniques and at others provide different information

-          A huge amount of results is based on no available experimental data and no references from other works to back up the statements.

-          PM10 is not fine particulate matter

-          Last paragraph of the abstract should be before

-          There is no paragraph stating what are the goals of the study in the introduction, the experimental section is too weak without supling the minimum information to the reader understand how the experimental setup and monitoring was done (there should be sub-sections: study area, sampling campaign (when, where); type of equipments, statistical treatment,…); there is no conclusions

-          Different units trough the paper

-          Lack of references everyone!

-          In the “Material and Methods”, the authors state that evaluate water soluble ions… no results are presented.

-          The authors say at some point that the sampling was done at 7 different times in a 3 moment period. In the discussion, they say that the sampling was done in different sites within the city. It is impossible to evaluate a paper like this.

-          What is “percentage of total number of days (%)” on table 2?

-          Comparison of results with WHO guideline values?

-          Is not possible to compare maximum concentrations with average concentrations like the authors do (lines 134-138)

-          Figures don’t have the years to which the sampling refers to

-          No information about how the data that is analysed was obtained

-          At some point, the authors provide analysis for other pollutants (SO2 and NO2) and parameters (wind and so on) – what data is that?

-          Figure 4 – use boxplot instead – it will provide meaningful information.

-          Figure 5 – ratios? How can the reader understand the PMx concentrations? What are the dots?

Author Response

Response to Reviewer 2 Comments

Point 1: The references are not updated with a total of 26 with only 1 reference from the last 7 years – 17 from 2000-2009 and 8 from 1999 or before. This shows how superficial the authors worked in this paper without providing a review of the latest work done in the study area – at least that should been done. For instance, with a short (1 min max) research, I found the following article:
Response 1: Thank you, we have quoted the latest articles in the relevant fields.
Point 2: The authors at some moments specify some pollutants and some techniques and at others provide different information.
Response 2: We removed pollutants that were not analyzed.
Point 3: A huge amount of results is based on no available experimental data and no references from other works to back up the statements.
Response 3: We have given the relevant data to the data source or documentation support.
Point 4: Last paragraph of the abstract should be before.
Response 4: Changed.
Point 5: There is no paragraph stating what are the goals of the study in the introduction, the experimental section is too weak without supling the minimum information to the reader understand how the experimental setup and monitoring was done (there should be sub-sections: study area, sampling campaign (when, where); type of equipments, statistical treatment,…); there is no conclusions.
Response 5: Thank you for your suggestions, we have made changes to the full text as you requested.
Point 6: Different units trough the paper
Response 6: We checked the full text of the unit and made a correction.
Point 7: Lack of references everyone
Response 7:We've added 7 new documents to support the content of the manuscript.
1. Liu, J., & Zhang, W. (2015). The influence of the environment and clothing on human exposure to ultraviolet light. PloS one 10(4): e0124758.
2. MEP. Technical Regulation on Ambient Air Quality Index(on trial): http:// kjs.mee.gov.cn/ hjbhbz /bzwb/ jcffbz/ 201203/ W020120410332725219541.pdf (accessed on 1 Jan 2016).
3. Royalty, T. M., Phillips, B. N., Dawson, K. W., Reed, R., Meskhidze, N., Petters, M. D. (2017). Aerosol properties observed in the subtropical north pacific boundary layer. Journal of Geophysical Research Atmospheres 122(18): 9990-10,012.
4. Shen, R., Klaus Schäfer, Jürgen Schnelle-Kreis, Shao, L., Norra, S., Kramar, U. , et al. (2018). Seasonal variability and source distribution of haze particles from a continuous one-year study in beijing. Atmospheric Pollution Research 9(4): 627-633.
5. Wang, G., Deng, T., Tan, H., Liu, X., Yang, H. (2016). Research on aerosol profiles and parameterization scheme in southeast china. Atmospheric Environment 140: 605-613.
6. Xia, Y., Tao, J., Zhang, L., Zhang, R., Li, S., Wu, Y., K Xiong, Z. (2017). Impact of size distributions of major chemical components in fine particles on light extinction in urban Guangzhou. Science of The Total Environment 240-247.
7. Zhang, J., Reid, J. S., Alfaro-Contreras, R., Xian, P. (2017). Has china been exporting less particulate air pollution over the past decade? Geophysical Research Letters 44(6):2941-2948.
Point 8: In the “Material and Methods”, the authors state that evaluate water soluble ions… no results are presented.
Response 8:We deleted this part because it didn't have a specific application.
Point 9: The authors say at some point that the sampling was done at 7 different times in a 3 moment period. In the discussion, they say that the sampling was done in different sites within the city. It is impossible to evaluate a paper like this.
Response 9 :We hope that sampling in different places will improve the universality of the data without being pseudo-representative of a point. We will update this section in the next study and thank you for your reminder.
Point 10: What is “percentage of total number of days (%)” on table 2?
Response 10 :We've labeled it after the table. * Number of days of major air pollutant per year as a proportion of total days.
Point 11: Is not possible to compare maximum concentrations with average concentrations like the authors do (lines 134-138)
Response 11:It is possible that we have already made a note.
Point 12: Figures don’t have the years to which the sampling refers to.
Response 12:We have explained -- the average monthly level from 2004-2010.
Point 13: No information about how the data that is analysed was obtained.
Response 13:We've given a description in the material and methods sections.
Point 14: At some point, the authors provide analysis for other pollutants (SO2 and NO2) and parameters (wind and so on) – what data is that?
Response 14:We've removed pollutants that didn't analyzed.
Point 15: Figure 5 – ratios? How can the reader understand the PMx concentrations? What are the dots?
Response 15:Changed.

Round 2

Reviewer 1 Report

I believe the authors clearly demonstrated effort in improving the manuscript. The results of this paper are interesting and the novelty of the article is evident. But I have still concerns about the poor presentation of the data and discussion as in many cases the reader will have hard time to understand the results. I have my new comments here and leave the final decision to the editor.   

Line 37: here “air quality” is meaningless as referred to a parameter that should be quantified. You can use a word like “the level of aerosol particles”

Table 1: The concentration unit for PM10 is still not provided in the table. It looks like you derived the correlation between API and C. In the table you should provide R2 values for those equations.

Table 2: Here is my previous comment: “Since it was previously mentioned that the measurements in this study were conducted in 2010, the data in Table 2 are brought from other sources. In this case, they should be cited.” I cannot see the reference for data provided in years other than 2010.

In response to my following comment on Table 2 “The maximum concentration for all the years except for the measured year are the same and 600. How can you justify the decrease happened only in 2010?” the authors responded “During this period, the API in Urumqi was set to no greater than 500”. Does this mean you obtained 536 ug/m3 using API=500 in the above-mentioned equations?  

The following comment is not responded “Also why no discussion on the other parameters such as annual average concentration is provided?”

My previous comment: “Line 151: “the proportion of PM10in the primary pollutants over the total number of days was also small over those two months.” Where are these proportions provided?” This sentence is still very confusing. It is the “percentage” not proportion. You should say that percentage versus the day. Currently it seems that you are referring to the concentration/number of day.

The same comment also applies to the recently added text, which has confusing wording and needed to be revised: “As a result, the ratio the number of days(what does it mean?) with PM10 as the primary pollutant to the total number of days is less than 70%, and from Figure 3 we found that this proportion decreased from January to March with March being the least number of days when PM10 being the primary pollutant, accounting for only 46.2%.

Line 231: “Based on the sound data obtained from the Urumqi weather station, the frequency, thickness and intensity of the ground inversion layer in Urumqi during the 2009 – 2010 period were analyzed. The results suggest that there is a close connection between the microphysical structure of PM and the degree of air pollution. Also, long-term temperature inversion has a great impact on the 234 microphysical structure of PM.” Where are those numbers? You cannot analyze and discuss data that are not presented and draw a conclusion out of it!

My previous comment is “Line 245: “Ozone and total oxidants (OX) show daily peaks in the mid-afternoon, i.e. the hours that experience higher solar radiation levels, and lower levels are experienced between 6 and 9 am local  time (daylight-saving time corrected in summer).” Where are these data (ozone and OX) you discussed?” You provided the reference for that but I cannot understand the relationship between those data and the measured data in this study. Why are the referenced data should be discussed here if they are not related to PM?

 “Comparing the ratios of different seasons suggested that the three ratios in winter are higher 278 than those in other seasons, which reflects the effect of winter inversion layer on the atmosphere. For 279 Winter (PM2.5/PM10), the maximum value appeared at 2pm, 9am, 10pm and 3 am, the minimum value 280 appeared at 6 pm, 8pm, 11am and 4am. Similar (how are they similar?!) to Winter (PM2.5/PM10), the maximum value of PM 281 mass concentrations ratio appeared at 3am, 12am and 10pm for Winter (PM1.0/PM10), the minimum 282 value appeared at 9 pm.”

The following added text is vague “However, aerosol mass concentration account for only 42.5% of the total mass concentration of PM10. but its mass was 42.5% of that of PM10.”

Author Response

Response to Reviewer 1 Comments

Point 1: Line 37: here “air quality” is meaningless as referred to a parameter that should be quantified. You can use a word like “the level of aerosol particles”

Response 1: Changed.

Point 2: Table 1: The concentration unit for PM10 is still not provided in the table. It looks like you derived the correlation between API and C. In the table you should provide R2 values for those equations.

Response 2: Changed.

Point 3: Table 2: Here is my previous comment: “Since it was previously mentioned that the measurements in this study were conducted in 2010, the data in Table 2 are brought from other sources. In this case, they should be cited.” I cannot see the reference for data provided in years other than 2010.

Response 3: The concentration of PM10 (from 2004-2009) was taken from the history data from the real-time Air quality index (AQI) monitoring network (http://aqicn.org/city/wulumuqi/)

Point 4: In response to my following comment on Table 2 “The maximum concentration for all the years except for the measured year are the same and 600. How can you justify the decrease happened only in 2010?” the authors responded “During this period, the API in Urumqi was set to no greater than 500”. Does this mean you obtained 536 ug/m3 using API=500 in the above-mentioned equations?  

Response 4: Thank you very much to the reviewers, we missed the definition of severe pollution in API (401-500), when the API exceeds 500, we calculate according to Formula C=API+100, when average daily maximum concentration will reach 600. See Table 1.

Point 5: My previous comment: “Line 151: “the proportion of PM10in the primary pollutants over the total number of days was also small over those two months.” Where are these proportions provided?” This sentence is still very confusing. It is the “percentage” not proportion. You should say that percentage versus the day. Currently it seems that you are referring to the concentration/number of day.

Response 5: We've replaced the proportion with percentage.

Point 6: The same comment also applies to the recently added text, which has confusing wording and needed to be revised: “As a result, the ratio the number of days (what does it mean?) with PM10 as the primary pollutant to the total number of days is less than 70%, and from Figure 3 we found that this proportion decreased from January to March with March being the least number of days when PM10 being the primary pollutant, accounting for only 46.2%.

Response 6: Changed. thank you for your reminder.

Point 7: Line 231: “Based on the sound data obtained from the Urumqi weather station, the frequency, thickness and intensity of the ground inversion layer in Urumqi during the 2009 – 2010 period were analyzed. The results suggest that there is a close connection between the microphysical structure of PM and the degree of air pollution. Also, long-term temperature inversion has a great impact on the 234 microphysical structure of PM.” Where are those numbers? You cannot analyze and discuss data that are not presented and draw a conclusion out of it!

Response 7: We thank the reviewer for his/her good suggestions. We have deleleted the sentences in the revision.

Point 8: My previous comment is “Line 245: “Ozone and total oxidants (OX) show daily peaks in the mid-afternoon, i.e. the hours that experience higher solar radiation levels, and lower levels are experienced between 6 and 9 am local  time (daylight-saving time corrected in summer).” Where are these data (ozone and OX) you discussed?” You provided the reference for that but I cannot understand the relationship between those data and the measured data in this study. Why are the referenced data should be discussed here if they are not related to PM?

Response 8: We have collected historical ozone data on the website http://aqidb.org/us/city/wulumuqi/

Point 9: “Comparing the ratios of different seasons suggested that the three ratios in winter are higher 278 than those in other seasons, which reflects the effect of winter inversion layer on the atmosphere. For 279 Winter (PM2.5/PM10), the maximum value appeared at 2pm, 9am, 10pm and 3 am, the minimum value 280 appeared at 6 pm, 8pm, 11am and 4am. Similar (how are they similar?!) to Winter (PM2.5/PM10), the maximum value of PM 281 mass concentrations ratio appeared at 3am, 12am and 10pm for Winter (PM1.0/PM10), the minimum 282 value appeared at 9 pm.”

Response 9: Changed.

Point 10: The following added text is vague “However, aerosol mass concentration account for only 42.5% of the total mass concentration of PM10. but its mass was 42.5% of that of PM10.”

Response 10: Changed.